POINT OF VIEW

# Open exploration

**Abstract** Arguments in support of open science tend to focus on confirmatory research practices. Here we argue that exploratory research should also be encouraged within the framework of open science. We lay out the benefits of 'open exploration' and propose two complementary ways to implement this with little infrastructural change.

WILLIAM HEDLEY THOMPSON[†*], JESSEY WRIGHT[†*] AND PATRICK G BISSETT[†*]

*For correspondence: william. thompson@stanford.edu (WHT); jessey.wright@gmail.com (JW); pbissett@stanford.edu (PGB)

†These authors contributed equally to this work

**Competing interests:** The authors declare that no competing interests exist.

## Introduction

The aim of open science is to increase the accessibility and transparency of scientific research by making changes to the way research is conducted (*Watson, 2015*; *Nosek et al., 2015*; *Levin and Leonelli, 2017*). To date many of the changes proposed and introduced to make research more open have been concerned with confirmatory research: examples include pre-registering studies (*Wagenmakers et al., 2012*), redefining statistical thresholds (*Benjamin et al., 2018*), increasing precision by focusing on statistical power (*Button et al., 2013*), and conducting replications (*Open Science Collaboration, 2015*).

These changes are all welcome, but we fear that this focus on confirmation gives the impression that confirmatory research is the sole part of the empirical process worth conducting openly, and that other parts – notably exploratory research – do not belong within the open-science movement. In this article we argue that that exploratory research does have a place in open science, but we do not believe open exploration will automatically follow from current initiatives that advocate open confirmatory practices (*Munafò et al., 2017*). Instead, exploratory research requires its own protocols and practices to bring it into the open. We start by outlining the problems that arise when exploration is closed and demonstrate that exploration is compatible with the ideals of open science. Thereafter, we outline concrete suggestions to integrate it into empirical practice.

## Moving from closed to open exploration

Data exploration can serve a variety of purposes, including: i) getting a holistic overview and understanding of the data; ii) generating a theory or hypothesis; iii) exploring degrees of freedom within the data. Sociological research detailing emotional labor (*Hochschild, 2012*), the discovery of penicillin (*Ligon, 2004*) and the training of AlphaGo (*Chen et al., 2018*) are a few examples of how exploratory research practices have been critical catalysts for discoveries throughout history. Routinely, advocates of exploratory research champion its value for science (*Behrens, 1997*; *Franklin, 2005*; *Hollenbeck and Wright, 2017*; *Jebb et al., 2017*; *Tukey, 1980*) and argue that it is insufficiently funded (*Aragon, 2011*; *Haufe, 2013*; *Rozin, 2009*; *Wagner and Alexander, 2013*). With the rise of 'data science', there has been an increased focus on data mining and extracting exploratory information and patterns (*Donoho, 2017*). In summary, while the exploration of data is necessary for scientific progress, it is not generally incentivized.

Exploration can be understood through degrees of freedom: the more degrees of freedom a researcher leverages, the more exploratory their work is. The leveraging of degrees of freedom can occur in study design (e.g., piloting experiments), during post-hoc analyses, and as an analysis strategy itself. Moreover, exploratory data analysis has been recognized as a practice distinct from confirmatory analyses since at least 1980 (*Tukey, 1980*). Following Tukey, we conceive of exploratory research as an interactive process in which a researcher adopts an attitude

of flexibility. This stands in a clear contrast with the characteristics of good confirmatory research. To put the contrast succinctly: productive exploratory research involves taking advantage of degrees of freedom, while good confirmatory practices involve limiting and reducing degrees of freedom.

While exploration is central to progress in many domains of science, it is rarely made public. The current blueprint for a scientific paper encourages the author to feign omniscience: the exact correct hypothesis was known a priori, only a small number of confirmatory statistical tests were run to address that hypothesis, and the tests came out exactly as predicted (*Grinnell, 2013*). This blueprint is further perpetuated by funding agencies and journals that reward these practices. In reality, before any confirmatory tests are run, there is often significant exploratory science. Therefore, perhaps in service of this omniscience myth, exploration is generally closed.

> **Productive exploratory research involves taking advantage of degrees of freedom, while good confirmatory practices involve limiting and reducing degrees of freedom.**

This poses several problems. First, by keeping exploration closed, there is little opportunity to improve one's own exploratory practice by studying the methods and strategies used by peers. This is a missed pedagogical opportunity because a big challenge for many young scientists is moving a research program from a blooming, buzzing confusion of infinite scientific possibilities to a specific question worth attempting to confirm. Second, because the standard for current papers is a tight, clean, linear, novel, and confirmatory narrative, researchers are encouraged to mischaracterize their exploratory science as confirmatory science (*Wagenmakers et al., 2012*). Third, many exploratory studies go unreported, contributing

to the 'file drawer problem' (*Rosenthal, 1979*). Fourth, a further consequence of 'closed exploration' is that a research group can waste time and resources on studies that others know will not work. Fifth, while closed exploration remains the norm, confirmation will be perceived as the only scientific process worth doing, thus discouraging researchers from doing good or thorough exploration. For these reasons, we advocate for open exploration.

In broad terms open exploration involves making all aspects of the exploratory research pipeline available for others to see what has been done, with what, and how. Such practices are compatible with current guidelines and initiatives that promote open research. Specifically, the transparency and openness promotion (TOP) guidelines were created to outline how research practices, broadly speaking, can be conducted openly (*Nosek et al., 2015*; https://cos.io/top/). Exploratory research can adhere to the sections of these guidelines that concern transparency of the data, code, research methods, design and analysis. The sections in the TOP guidelines that are not relevant for open exploration are pre-registration (as this demarcates confirmatory from exploratory research) and replication (as this is a type of confirmatory research). Finally, in advocating for and thinking about open exploration, it is critical to recognize that open exploration complements confirmatory research, rather than somehow being in opposition to it.

## Proposed implementations of open exploration

We propose two concrete ways to instantiate open exploration: i) include the exploratory science as a distinct section in published papers; ii) place exploratory analyses (regardless of the outcome) on citable public repositories. Before elaborating on these proposals, we present four important additional details. First, we recognize that these solutions will not rectify the entire problem. We offer them as concrete steps forward that are low cost. Second, our proposals aim to promote exploration that does not currently see the light of day. Notably, there is some exploratory research that is already in the open, such as post hoc analyses and fully exploratory studies – see, for example, the 'exploratory reports' format adopted by the journal *Cortex*. Third, we admit that these proposals are not necessarily novel but they are stated here to

encourage their adoption. Fourth, our experience is primarily in psychology and neuroscience, so our proposals may be best suited to these and related fields. However, we hope that they may be of value to other disciplines that are facing similar challenges as there is nothing discipline-specific about them.

One way to implement the first suggestion would be to include separate sections in scientific papers for exploration. To fit this neatly into current structures, this may involve an exploratory methods section and an exploratory results section. Regardless of the specific implementation, such section(s) would give researchers an opportunity to provide details about pilot studies, initial but incorrect hypotheses, and descriptive and exploratory data analyses that were conducted prior to the confirmatory part of the study. Such section(s) do not need to focus on, if at all, inferential statistical results or power: instead, they should describe the exploratory work that was done and how it led to the hypothesis generation, methods refinement, or any additional insight for the confirmatory research. The subsequent (confirmatory) methods and results sections will then be informed by the exploratory results reported in the paper and would be held to the strict standards of confirmatory science. This re-imagining of the scientific paper will demonstrate that exploratory research is as an important part of the scientific process, instead of discouraging it or pretending it does not occur.

> **This re-imagining of the scientific paper will demonstrate that exploratory research is as an important part of the scientific process, instead of discouraging it or pretending it does not occur.**

Adopting the first proposal will only make the tip of the exploratory iceberg open. To address the file drawer problem and avoid redundant work by researchers, other exploratory work that is currently considered to be 'non-publishable' must also be placed in the open. However, we do not want to flood the scientific literature with exploratory work that does not necessarily warrant a publication, so we propose that researchers place their exploratory analysis – ideally including the code, data (if it is shareable), and descriptions of the study – on a repository with a citable DOI. (Suitable repositories include figshare, Github and osf.io, and these can be integrated with DOIs generated from Zenodo or protocols.io). There are already some instances where exploratory analyses are shared in this way (see, for example, *Konkiel, 2016*). For wider adoption of this suggestion, and to ease readability of exploratory analyses in repositories, developing a standardized template to describe the exploratory repository could be beneficial.

## Challenges and conclusions

We have outlined two proposals that implement open exploration with little infrastructural change. Together they would help to curtail the problems of closed exploration outlined previously: for example, by giving a place to exploration in journal articles, more parts of the scientific process will be conducted in the open, which would help to dispel the omniscience myth, and would also generate material that can be used for future investigations into good exploratory practices. Moreover, open exploration may make it possible to conduct meta-exploratory analyses that identifies both successful and under-exploited places in the parameter space to guide future work. This could be done by, for example, reviewing exploratory analysis repositories and presenting a summary of the parameter configurations that have been explored (noting where researcher degrees of freedom appear to make a difference) and the null and positive findings contained in the analyses.

We would also like to address some concerns that researchers might have about our proposals. First, some researchers may be hesitant to publicly display exploratory work that is incomplete, such as rough or messy code. However, sharing any code, even messy or rough code, is better than having no code accompanying an article: indeed, it has been shown that sharing any code at all increases scientific engagement with articles, as measured by citations, in image processing research (*Vandewalle, 2012*). See *Gorgolewski and Poldrack (2016)* for more advice on sharing code (and data). Second, it is important that people do not overlook that null exploratory findings could be type-II errors (that is, false negatives). Third,

there is also a possibility of more type-I errors (that is, false positives) in open exploration. However, as more exploratory analyses are shared, it may become possible to discriminate between findings worth confirming (be they negative or positive) and type-I or type-II errors due to factors such as a poor experimental design. At present, news of negative results encountered in unpublished work due to poor experimental design can only be spread through word-of-mouth, dissuading others from pursuing this line of research. Open exploration offers the possibility for these results to be evaluated. In sum, it is crucial that exploratory work should be seen as a guide for future work, not as definitively confirmed hypothesis.

Finally, there is the concern that positive or negative exploratory results that have not been confirmed may be adopted as facts or falsehoods by the public. However, misleading information already exists and gets proliferated as facts (see, for example, *Nordenstedt and Rosling, 2016*). With greater appreciation of exploration, the distinction between researchers producing tentative exploratory results and hypothesis-driven research will become more apparent to both scientists and a wider audience.

To conclude, open science has focused on confirmatory science and neglected exploratory science. Exploration exists, has scientific value, and is mostly closed. We propose open exploration, which involves bringing exploratory science out of the shadows into scientific papers and public repositories, and in doing so unlocks the potential of this essential part of the scientific process.

## Acknowledgements
We thank Russell Poldrack and James Mac Shine for helpful comments and discussions.

**William Hedley Thompson** is in the Department of Psychology, Stanford University, Stanford, United States, and the Department of Clinical Neuroscience, Karolinska Institute, Stockholm, Sweden
william.thompson@stanford.edu
https://orcid.org/0000-0002-0533-6035

**Jessey Wright** is in the Department of Psychology and the Department of Philosophy, Stanford University, Stanford United States
jessey.wright@gmail.com

**Patrick G Bissett** is in the Department of Psychology, Stanford University, Stanford United States
pbissett@stanford.edu

**Author contributions:** William Hedley Thompson, Jessey Wright, Patrick G Bissett, Conceptualization, Writing - original draft, Writing - review and editing

**Competing interests:** The authors declare that no competing interests exist.

## Funding

| Funder | Grant reference number | Author |
|---|---|---|
| Knut och Alice Wallenbergs Stiftelse | 2016.0473 | William Hedley Thompson |
| National Institute of Mental Health | R01MH117772 | Patrick G Bissett |
| National Science Foundation | 1655839 | Jessey Wright |
| Social Sciences and Humanities Research Council of Canada | 756-2017-0766-SSHRC | Jessey Wright |
| National Institute on Drug Abuse | UH3DA041713 | Patrick G Bissett |

The funders had no role in study design, data collection and interpretation, or the decision to submit the work for publication.

**Decision letter and Author response**
Decision letter https://doi.org/10.7554/eLife.52157.sa1
Author response https://doi.org/10.7554/eLife.52157.sa2

## Additional files

### Data availability
No data is used in this article.

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
