## [Decision Letter]

Thank you for submitting your article "Open Exploration" for consideration by *eLife*. Your article has been reviewed by three peer reviewers, and the evaluation has been overseen by myself (Peter Rodgers) as the *eLife* Features Editor. The following individuals involved in review of your submission have agreed to reveal their identity: Micheal de Barra (Reviewer #2).

As you will see, the reviewers #1 and #3 were somewhat negative about the manuscript, whereas reviewer #2 was more positive (though he also had concerns). I would be willing to consider a revised version of the manuscript, but it would require major revisions, and you may prefer to seek publication elsewhere.

Reviewer #1:

The thesis of this article is stately concisely in the abstract and conclusion. Abstract: "Science is in the midst of an open science revolution that focuses on conﬁrmatory research practices. This has the unfortunate byproduct of neglecting or even dissuaded [sic] good exploratory research." Conclusion: "...open science has focused on conﬁrmatory science and neglected exploratory science."

I don't agree. Of course this is an opinion piece, so I don't have to agree, but I think the authors fail to provide evidence that a problem exists - or put better, that the problem is as they state it to be. I summarize my main disagreements with reference to statements in the text.

1) "Science is in the midst of an open science revolution that emphasizes reproducible research, data sharing, standardization, and open access to scientiﬁc articles." To me this is a misstatement of what open science if about. Open science in my view emphasizes data sharing and open access. Reproducibility and standardization are extremely important but I wouldn't put them under the same heading. I never thought I'd find myself quoting Wikipedia in a scholarly context, but here's how they define it: "Open science is the movement to make scientific research (including publications, data, physical samples, and software) and its dissemination accessible to all levels of an inquiring society, amateur or professional.... It encompasses practices such as publishing open research, campaigning for open access, encouraging scientists to practice open notebook science, and generally making it easier to publish and communicate scientific knowledge."

2) "Current prescriptions on how to integrate openness into empirical scientiﬁc practice have focussed on the conﬁrmatory stages.... This focus gives the impression that conﬁrmatory research is the sole part of the empirical process worth pursuing and disseminating." Not in my world, which is biology, including some experience in cell biology, molecular biology, biochemistry, anatomy, physiology, developmental biology and neurobiology. I see that the authors are all from a psychology department so their world is surely different from mine. If the statement is true in their world, they need to (a) make that clear and (b) provide more evidence for this rather irresponsibly broad accusation.

3) "...because the standard for current papers is a tight, clean, linear, and conﬁrmatory narrative, this encourages researchers to mischaracterize their exploratory science as conﬁrmatory science." Publication pressures are real and pernicious, especially in the for-profit top-tier journals. In my world, though, I don't see the pressures as regularly emphasizing confirmation. If anything, the pressure is for novelty, even when the foundation may be a bit shaky. So again, if psychology plays by a different set of rules, the authors need to specify who they are addressing and accusing.

4) "...include the exploratory science as a distinct section in published papers..." I think this is a poor idea and would actually stigmatize the exploratory aspect. However, I don't dispute the authors' right to make the suggestion.

5) "To move towards fully solving problems regarding the ﬁle drawer and avoiding redundant work by researchers, the non-publishable exploration has to be placed in the open too." There is a lot of merit to this idea, but it has some drawbacks that need to be described. One is that if "exploratory" research comes to be seen as "whatever is stuck in your file drawer" it becomes difficult for readers to know whether the negative result is real (in which case they can avoid redundancy) or whether it reflects poor experimental design and execution (in which case they might be dissuaded from a productive line of research).

Reviewer #2:

I enjoyed reading this manuscript and am in agreement with many of the points raised about the importance of exploratory and descriptive research in contemporary psychology. In particular the paragraph on the motivation for making exploratory research open is excellent.

A few points to consider:

The authors write "[Open science] has the unfortunate byproduct of neglecting or even dissuaded good exploratory research." Whatever the truth of this empirical claim, open science and exploratory research need not and ought not be seen in such a zero-sum light. The open science movement have made the distinction between exploratory and confirmatory research much clearer to researchers - our statistical tests only make sense in conditions where there are few degrees of freedom. But our new-found clarity about requirements of confirmatory research should liberate and legitimise exploratory and descriptive research since it no longer needs to "pretend" to be confirmatory (p-values, consistent results etc). This point is well made by Frankenhuis and Nettle 2018 (http://doi.org/10.1177/1745691618767878, see also other articles in this special edition). Useful to stress the compatibility of open science and exploratory research up-front rather than on page 3?

I am a little concerned that the authors 'proposals' are either a bit arbitrary or already reasonably common. What’s the difference between a separate "exploratory" methods / results and a pilot study? Many articles already clearly label confirmatory results (which where preregistered) and exploratory results (which were not preregistered). Figshare and OSF are already used to share datasets that may not warrant a publication.

While the authors mention that inferential statistics will play a reduced role in exploratory work, the language seems to assume null hypothesis significance testing: see, e.g., references to "null finding", "failed hypothesis" or "success" in exploratory research). While I don't think this is the intention of the authors, such language gives the impression that exploratory research is just 'failed' confirmatory research. Moreover, it's likely that statisticians have written about good statistical approaches and NHST in exploratory work: is there anything useful they cite or describe here? (splitting datasets for explore and confirm, e.g.)

Relatedly, it would be useful to have more discussion of the kinds of exploratory research that the authors wish to create more space and prestige for. Paul Rozin's 2009 paper "What Kind of Empirical Research Should We Publish, Fund, and Reward?: A Different Perspective" provides some interesting examples of valuable non-confirmatory research. Tinbergen's 1963 comment that "on its haste to step into the twentieth century and to become a respectable science, Psychology skipped through the preliminary descriptive stage that other natural sciences had gone through" also springs to mind. Have the authors any thoughts on this?

Reviewer #3:

The premise of this provocative article is mistaken: the open science "revolution" does not imply an increased bias towards confirmatory science.

As Wikipedia says, "Open science is the movement to make scientific research (including publications, data, physical samples, and software) and its dissemination accessible to all levels of an inquiring society, amateur or professional." There is nothing in that statement that implies a bias against publishing exploratory work.

Open science practices already enable what the authors describe as "exploratory data analysis", by encouraging (or insisting) that scientists make data available when they publish results based on it. This is the proper time to release data, as uncurated data is worse than none at all, and the publication process is the best tool we have for judging whether data is meaningful.

The trend toward confirmatory science versus more exploratory work originates primarily in the practices of funding agencies and journals, and especially their reviewer communities. These communities are unduly influenced by the simplistic example of hypothesis testing commonly practiced in the biological and pharmaceutical sciences.

The authors introduce the term "closed exploration" to mean hiding the thought process that led to the hypothesis being tested. They describe well the disadvantages of hiding this thought process, and focusing only on hypothesis testing (called by others "hypothesis-driven" research).

However, this is not a new insight; for earlier discussions see Aragon "Thinking Outside The Box: Fostering Innovation and Non-Hypothesis Driven Research at NIH" (2011), Haufe "Why do funding agencies favor hypothesis testing?" (2013), O'Malley et al "Philosophies of Funding" (2009), and McKnight "The straight-jacket of hypothesis driven research" (2015).

The proper remedy for this tendency to "feign omniscience" is for funding agencies and journals to recognize that the story science should tell is as much how we arrive at a good hypothesis, as how we test it. If that were more widely appreciated, material made public through open science channels would reflect that improved understanding of what science is.

Part of the authors' prescription for addressing the tendency to feigned omniscience, is that journals should add "exploratory" methods and "exploratory" results sections to the conventional Methods and Results. This is fine as far as it goes; the structure of Methods and Results is ill adapted to telling a story of discovery, and there are many ways of improving upon it. Relaxing this rigid structure would improve journals that insist upon it.

The authors place considerable importance on what they call the "file drawer problem", which they describe as the accumulation of "unreported exploration". The novel aspect of their remedy for the present bias towards confirmatory science is to publish in online repositories incomplete stories - what they call "exploratory analyses (regardless of success)".

That prescription strikes me as naive, with regard to the social practice of science. Exploratory work is hard labor not valuable on its own - rather like preparing the soil for a garden. No one will be motivated to share that. Furthermore, exploratory work that has not yet led to a clear result is not really a tellable story. If the story is ready to tell, scientists are strongly motivated to tell it. There would be little value in fishing through accounts of other people's explorations, except to know what they have been trying to do - which no one would willingly submit to.

In summary, the authors need to think harder about the underlying causes of bias towards confirmatory research. The open science movement is not to blame, nor is it clear that it affords a remedy.

---

## [Author Response]

[We repeat the reviewers’ points here in italic, and include our replies point by point, as well as a description of the changes made, in Roman.]

Reviewer 1:1) “Science is in the midst of an open science revolution that emphasizes reproducible research, data sharing, standardization, and open access to scientiﬁc articles.” To me this is a misstatement of what open science if about. Open science in my view emphasizes data sharing and open access. Reproducibility and standardization are extremely important but I wouldn’t put them under the same heading. I never thought I’d find myself quoting Wikipedia in a scholarly context, but here’s how they define it: “Open science is the movement to make scientific research (including publications, data, physical samples, and software) and its dissemination accessible to all levels of an inquiring society, amateur or professional…. It encompasses practices such as publishing open research, campaigning for open access, encouraging scientists to practice open notebook science, and generally making it easier to publish and communicate scientific knowledge.”

Reply: Firstly, we have revised our first two paragraphs to make a clear distinction between the ideals of open science and the policies and practices that implement it. Further, we have tried to be explicit about where/how exploration is being marginalized by these implementations.

Secondly, the reviewer is saying the reproducibility and standardization are not parts of open science by citing Wikipedia. Here we disagree with the reviewer. Academic sources, such as Nosek et al 2015 and the transparency and openness promotion (TOP) guidelines talk at length about reproducibility as part of an open research environment. Furthermore, our first citation (Levin & Lenolli 2017) talks about the broadness of open science. It contains the following text:

“the RCUK [Research Councils UK] Policy on Open Access poses challenges for researchers because papers reference to data sets, software, models, instruments, protocols, and know-how that should also be shared for the contents of papers to be intelligible and reproducible” (Levin & Lenolli 2017 pg 282-283)

We favor the peer reviewed academic definitions and policies of open science to the Wikipedia definition. As stated above, we have tried to be more precise with how we discuss open science. Together, we hope this alleviates the reviewer’s concerns.

2) “Current prescriptions on how to integrate openness into empirical scientiﬁc practice have focussed on the conﬁrmatory stages…. This focus gives the impression that conﬁrmatory research is the sole part of the empirical process worth pursuing and disseminating.” Not in my world, which is biology, including some experience in cell biology, molecular biology, biochemistry, anatomy, physiology, developmental biology and neurobiology. I see that the authors are all from a psychology department so their world is surely different from mine. If the statement is true in their world, they need to (a) make that clear and (b) provide more evidence for this rather irresponsibly broad accusation.

Reply: Please see responses to point 1 above. Additionally, we now state:

We *fear that this focus* on confirmation gives the impression that this type of research is the sole part of the empirical process worth conducting openly, and so the only stage of research worth pursuing, receiving credit for having done, and disseminating. [emphasis added]

We have also included a disclaimer about our background and made it explicitly clear that we are not saying our discussion is a solution that necessarily applies to all fields:

Fourth, our experience is primarily in psychology and neuroscience, so our proposals may be best suited to these and related fields. However, we hope that they may be of value to other disciplines that are facing similar challenges as there is nothing discipline-specific about them.

3) “…because the standard for current papers is a tight, clean, linear, and conﬁrmatory narrative, this encourages researchers to mischaracterize their exploratory science as conﬁrmatory science.” Publication pressures are real and pernicious, especially in the for-profit top-tier journals. In my world, though, I don’t see the pressures as regularly emphasizing confirmation. If anything, the pressure is for novelty, even when the foundation may be a bit shaky. So again, if psychology plays by a different set of rules, the authors need to specify who they are addressing and accusing.

Reply: We agree with Reviewer 1 that novelty drives publication pressure. Given this, we have added “novel” to the aforementioned sentence. However, we do not see novelty as necessarily linked to exploration and not confirmation. Indeed, from our perspective, top-tier journals want all of the things that we list (“…tight, clean, linear, novel, and confirmatory…”). In other words, a novel finding without confirmation is just a hunch that needs more research before being suitable for a top-tier journal.

However, we understand that our perspective may apply better to certain fields than others. This recognition is another reason why we have added the clarification about our disciplinary situation noted in reply to reviewer 1 point 2.

4) “…include the exploratory science as a distinct section in published papers…” I think this is a poor idea and would actually stigmatize the exploratory aspect. However, I don’t dispute the authors’ right to make the suggestion.

Reply: We thank Reviewer 1 for their perspective on this point. We also appreciate that Reviewer 1 is open to our suggestion, even if they think it is a poor idea. We hope that this work starts a conversation about whether science can be improved by encouraging the open dissemination of exploratory science. We also hope that this conversation will yield new and perhaps better ideas for how to implement this.

Adding distinct section(s) for exploratory science in published papers results from two convergent considerations. First, given the central role that publications have in scientific discourse, we feel that including exploratory work within published papers is an essential part of opening exploratory science. Second, we think that it is essential that exploratory science is distinguished from confirmatory science, as they necessitate different statistical procedure and they should be evaluated differently. Given these two considerations, we feel that adding distinct sections in published papers has a potentially useful role in opening exploratory science. However, these details are beyond the scope of our short letter, and so we have not included any changes.

5) “To move towards fully solving problems regarding the ﬁle drawer and avoiding redundant work by researchers, the non-publishable exploration has to be placed in the open too.” There is a lot of merit to this idea, but it has some drawbacks that need to be described. One is that if “exploratory” research comes to be seen as “whatever is stuck in your file drawer” it becomes difficult for readers to know whether the negative result is real (in which case they can avoid redundancy) or whether it reflects poor experimental design and execution (in which case they might be dissuaded from a productive line of research).

Reply: We agree with the reviewer that this is a potential problem. We briefly touch upon this issue in the paragraph that begins with: “We have outlined how open exploration can be implemented but there are a number of concerns researchers may have”. We stated that it was important for researchers to consider whether null exploratory findings were type II errors. We have expanded upon this, by now stating:

Second, it is important that people do not overlook that null exploratory findings could be type-II errors. Third, there is also a possibility of more type-I errors existing in open exploration which is a potential problem regarding the interpretation of exploratory findings. As more exploratory analyses are shared, it will become possible to detect when type-I and type-II errors are occurring due to additional factors such as a poor experimental design. At present, a closed exploratory type-II error due to poor experimental design could be spread through word-of-mouth at labs and conferences dissuading others to pursue this line of research. Open exploration offers the possibility for these results to be evaluated. In sum, it is crucial that exploratory work should be seen as a guide for future work, not as definitively confirmed hypothesis.

We hope we have managed to address the reviewer’s concern and at the same time show how the concern is currently a problem with “closed exploration”, and how “open exploration”, if results are treated properly, helps alleviate this concern.

Reviewer 2:The authors write “[Open science] has the unfortunate byproduct of neglecting or even dissuaded good exploratory research.” Whatever the truth of this empirical claim, open science and exploratory research need not and ought not be seen in such a zero-sum light. The open science movement have made the distinction between exploratory and confirmatory research much clearer to researchers - our statistical tests only make sense in conditions where there are few degrees of freedom. But our new-found clarity about requirements of confirmatory research should liberate and legitimise exploratory and descriptive research since it no longer needs to “pretend” to be confirmatory (p-values, consistent results etc). This point is well made by Frankenhuis and Nettle 2018 (http://doi.org/10.1177/1745691618767878, see also other articles in this special edition). Useful to stress the compatibility of open science and exploratory research up-front rather than on page 3?

Reply: We strongly agree with Reviewer 2 that open science and exploratory research need not and ought not be seen in a zero-sum light. This point was a primary impetus for us to write this letter. We have made multiple changes to try to make this more clear. We would like to point out one salient example from the first page, in which we now say:

“Here we argue that exploration is compatible with open science and open exploration is beneficial.”

We also add the following point, suggested by Reviewer 2, about the implications of the Frankenhuis and Nettle (2018): “Indeed, open exploration should provide a clearer distinction between exploration and confirmation that can help liberate and legitimize exploration (Frankenhais & Nettle, 2018).”

I am a little concerned that the authors ‘proposals’ are either a bit arbitrary or already reasonably common. What’s the difference between a separate “exploratory” methods / results and a pilot study? Many articles already clearly label confirmatory results (which where preregistered) and exploratory results (which were not preregistered). Figshare and OSF are already used to share datasets that may not warrant a publication.

Reply: First, we did not necessarily intend to claim these proposals were necessarily novel. We apologise if it read this way. We have explicitly added the following statement:

Second, because the standard for current papers is a tight, clean, linear, novel, and confirmatory narrative, this encourages researchers to mischaracterize their exploratory science as confirmatory science (Wagenmakers et al., 2012).

Second, despite the solutions not necessarily being novel, we hope that highlighting them may encourage their wider adoption. We see a parallel to the literature on preprints, which has many articles championing their benefits even if presented solutions are not novel. We think that championing some actionable implementations of Open Exploration is an important addition to this paper.

To our knowledge, the adoption of our proposed solutions is not commonplace. The reviewer is correct that many datasets are on Figshare and OSF. However, we are not merely suggesting that datasets be put online, but the entire analysis of what was explored on the dataset be shared. Indeed, it was challenging to find a single good example on Figshare or anywhere else that instantiated our vision of sharing their exploratory work in a repository. We now reference this example from Figshare. The text now reads:

Our second proposed solution would be for researchers to place their exploratory analysis on a repository (e.g. Github or Figshare). These repositories can have a citable DOI (e.g. via Zenodo or protocols.io). Such DOIs have been successfully implemented for software, datasets, and journal articles, allowing them to be credited by others (Peters, Kraker, Lex, Gumpenberger, & Gorraiz, 2017). There are already some instances where exploratory analyses are shared in this way (e.g. Konkiel (2016)). However, for wider adoption this suggestion requires researchers to consider presenting their analyses in a more organized fashion to maximise the utility of these repositories.

More generally, we feel that pre-confirmatory study exploration can be much broader than pilots, making our suggestion more than just detailing pilot studies. This point is made in our paragraph discussing the relationship between exploration and degrees of freedom (“Exploration can be understood…) and elsewhere.

While the authors mention that inferential statistics will play a reduced role in exploratory work, the language seems to assume null hypothesis significance testing: see, e.g., references to “null finding”, “failed hypothesis” or “success” in exploratory research). While I don’t think this is the intention of the authors, such language gives the impression that exploratory research is just ‘failed’ confirmatory research. Moreover, it’s likely that statisticians have written about good statistical approaches and NHST in exploratory work: is there anything useful they cite or describe here? (splitting datasets for explore and confirm, e.g.)

Reply: We agree that our language inappropriately assumes NHST and we have amended our language. Specifically, we changed “failed hypothesis” to “incorrect hypothesis” and “success” to “outcome” when talking about exploratory results. We have not changed the two instances of “null” findings, because this language could be used in any hypothesis testing framework. Further they only occur in the paragraphs either proposing additional ideas or talking about wider implications (e.g. saying “null exploratory findings” does not, to us, assume NHST given preceding text).

Relatedly, it would be useful to have more discussion of the kinds of exploratory research that the authors wish to create more space and prestige for. Paul Rozin’s 2009 paper “What Kind of Empirical Research Should We Publish, Fund, and Reward?: A Different Perspective” provides some interesting examples of valuable non-confirmatory research. Tinbergen’s 1963 comment that “on its haste to step into the twentieth century and to become a respectable science, Psychology skipped through the preliminary descriptive stage that other natural sciences had gone through” also springs to mind. Have the authors any thoughts on this?

REPLY: Our answer to this is first get all exploratory research in the open, then we can truly see which we practice we should encourage and give more prestige too. We already had a sentence in the paragraph relating to the benefits of open exploration that stated:

“Conducting exploration in the open will also generate material that can be used for future investigations into good exploratory practices.”

Furthermore, until exploration is fully in the open, it is unable to state an evidence-based opinion about this question so we do not think it is our place to advocate for specific types of exploration at present. However, also in conjunction to concerns about reviewer 3, we have brought up the literature regarding funding about non-confirmatory/exploratory research and state:

Routinely, advocates of exploratory research champion its value for science (Behrens, 1997; Franklin, 2005; Hollenbeck & Wright, 2017; Jebb et al., 2017; Tukey, 1980) and argue that it is insufficiently funded (Aragon, 2011; Haufe, 2013; Rozin, 2009; Wagner & Alexander, 2013)

This text now includes the Rozin paper the reviewer suggested. However while we like the comment by Tinbergen, we were unable to find a natural place for to place this reference in the article as we do not want to make the article solely about psychology. We have however added a number of additional references (see 0.2).

Reviewer 3:The premise of this provocative article is mistaken: the open science “revolution” does not imply an increased bias towards confirmatory science. As Wikipedia says, “Open science is the movement to make scientific research (including publications, data, physical samples, and software) and its dissemination accessible to all levels of an inquiring society, amateur or professional.” There is nothing in that statement that implies a bias against publishing exploratory work.Open science practices already enable what the authors describe as “exploratory data analysis”, by encouraging (or insisting) that scientists make data available when they publish results based on it. This is the proper time to release data, as uncurated data is worse than none at all, and the publication process is the best tool we have for judging whether data is meaningful.

Reply: These points have been addressed in point 1 raised by reviewer 1. In brief, we have tried to be clearer in our first two introductory paragraphs where we see current initiatives (not ideals) of open science are adapted to confirmatory, not exploratory research.

The trend toward confirmatory science versus more exploratory work originates primarily in the practices of funding agencies and journals, and especially their reviewer communities. These communities are unduly influenced by the simplistic example of hypothesis testing commonly practiced in the biological and pharmaceutical sciences.

Reply: We agree with Reviewer 3 that funding agencies and journals may play a significant role in emphasizing the tight, clean, linear narrative of confirmatory science. We make this explicit, and credit this point to the reviewer, by saying now by saying:

The current blueprint for a scientific paper encourages the author to feign omniscience: the exact correct hypothesis was known a priori, only a small number of confirmatory statistical tests were run to address that hypothesis, and the tests came out exactly as predicted (Grinnell, 2013). As pointed out by a reviewer, this blueprint is further perpetuated by funding agencies and journals that reward these practices. In reality, before any confirmatory tests are run, there is often significant exploratory science.

The authors introduce the term “closed exploration” to mean hiding the thought process that led to the hypothesis being tested. They describe well the disadvantages of hiding this thought process, and focusing only on hypothesis testing (called by others “hypothesis-driven” research). However, this is not a new insight; for earlier discussions see Aragon “Thinking Outside The Box: Fostering Innovation and Non-Hypothesis Driven Research at NIH” (2011), Haufe “Why do funding agencies favor hypothesis testing?” (2013), O’Malley et al “Philosophies of Funding” (2009), and McKnight “The straight-jacket of hypothesis driven research” (2015).

Reply: We do not intend for this article to focus on funding in science. As the Reviewer points out, this has been the focus of multiple recent papers. We intend for our article to focus on open exploration, which to our knowledge is not addressed by the suggested papers. However, we acknowledge that exploratory research and funding has a long history together. Thus in our discussion of exploration we add the following sentence:

Routinely, advocates of exploratory research champion its value for science (Behrens, 1997; Franklin, 2005; Hollenbeck & Wright, 2017; Jebb et al., 2017; Tukey, 1980) and argue that it is insufficiently funded (Aragon, 2011; Haufe, 2013; Rozin, 2009; Wagner & Alexander, 2013).

Additionally, we address the role of funding in response to reviewer 3 point 2. However, we do not intend for this article to focus on funding in science.

The proper remedy for this tendency to “feign omniscience” is for funding agencies and journals to recognize that the story science should tell is as much how we arrive at a good hypothesis, as how we test it. If that were more widely appreciated, material made public through open science channels would reflect that improved understanding of what science is. Part of the authors’ prescription for addressing the tendency to feigned omniscience, is that journals should add “exploratory” methods and “exploratory” results sections to the conventional Methods and Results. This is fine as far as it goes; the structure of Methods and Results is ill adapted to telling a story of discovery, and there are many ways of improving upon it. Relaxing this rigid structure would improve journals that insist upon it. The authors place considerable importance on what they call the “file drawer problem”, which they describe as the accumulation of “unreported exploration”. The novel aspect of their remedy for the present bias towards confirmatory science is to publish in online repositories incomplete stories - what they call “exploratory analyses (regardless of success)”. That prescription strikes me as naive, with regard to the social practice of science. Exploratory work is hard labor not valuable on its own - rather like preparing the soil for a garden. No one will be motivated to share that. Furthermore, exploratory work that has not yet led to a clear result is not really a tellable story. If the story is ready to tell, scientists are strongly motivated to tell it. There would be little value in fishing through accounts of other people’s explorations, except to know what they have been trying to do - which no one would willingly submit to.

Reply: We respectfully disagree with the reviewer. Firstly, there is evidence that exploration is sometimes openly shared. We have now provide a citation to an exploratory analysis on Figshare (see response to reviewer 2 point 2). Second, scientific communication need not include storytelling. Indeed, there is a journal Matters dedicated to observational research where there is no story (see: https://sciencematters.io/articles/140385074186). Third, we believe that the open science movement involves encouraging a change in motivations and actions. Even if few researchers are currently motivated to share exploratory science, we are trying to propose solutions that may change their motivations and actions. Fourth, storytelling is a rhetorical tool to make science engaging, but the goal of science is not to tell a story. Science is about understanding and explaining the world. We think the opening exploration can help to understand and explain the world, even if it does not lend itself to story. Fifth, we explicitly put forward the need to systematize sharing exploratory research in such a way that dredging through what other people have done is not too hard to do. The reviewer may not want to participate in open exploration or contribute their research in this way, that is their prerogative. But we have provided evidence why we do not consider that this opinion is not representative of science as a whole.

**References**

Aragon, R. (2011). Thinking Outside the Box: Fostering Innovation and Non-Hypothesis-Driven Research at NIH. *Science Translational Medicine, 3*(70). https://doi.org/10.1126/scitranslmed.3001742

Banks, G. C., Field, J. G., Oswald, F. L., O’Boyle, E. H., Landis, R. S., Rupp, D. E., & Rogelberg, S. G. (2019). Answers to 18 Questions About Open Science Practices. *Journal of Business and Psychology, 34*(3), 257–270. https://doi.org/10.1007/s10869-018-9547-8

Behrens, J. T. (1997). Principles and Procedures of Exploratory Data Analysis. *Psychological Methods, 2*(2), 131–160. https://doi.org/10.1037/1082-989X.2.2.131

Benjamin, D. J., Berger, J. O., Johannesson, M., Nosek, B. A., Wagenmakers, E.-J., Berk, R., … Johnson, V. E. (2018). Redefine statistical significance. *Nature Human Behaviour, 2*(1), 6–10. https://doi.org/10.1038/s41562-017-0189-z

Button, K. S., Ioannidis, J. P., Mokrysz, C., Nosek, B. A., Flint, J., Robinson, E. S., & Munafò, M. R. (2013). Power failure: Why small sample size undermines the reliability of neuroscience. *Nature Reviews Neuroscience, 14*(5), 365–376. https://doi.org/10.1038/nrn3475

Frankenhuis, W. E., & Nettle, D. (2018). Open Science Is Liberating and Can Foster Creativity. *Perspectives on Psychological Science, 13*(4), 439–447. https://doi.org/10.1177/1745691618767878

Franklin, L. R. (2005). Exploratory experiments. *Philosophy of Science, 72*(5), 888–899. https://doi.org/10.1086/508117

Grinnell, F. (2013). Research integrity and everyday practice of science. *Science and Engineering Ethics, 9*(3), 685–701. https://doi.org/10.1007/s11948-012-9376-5

Haufe, C. (2013). Why do funding agencies favor hypothesis testing? *Studies in History and Philosophy of Science Part A, 44*(3), 363–374. https://doi.org/10.1016/j.shpsa.2013.05.002

Hollenbeck, J. R., & Wright, P. M. (2017). Harking, Sharking, and Tharking: Making the Case for Post Hoc Analysis of Scientific Data. *Journal of Management, 43*(1), 5–18. https://doi.org/10.1177/0149206316679487

Jebb, A. T., Parrigon, S., & Woo, S. E. (2017). Exploratory data analysis as a foundation of inductive research. *Human Resource Management Review, 27*(2), 265–276. https://doi.org/10.1016/j.hrmr.2016.08.003

Konkiel, S. (2016). *What kinds of research are cited quickly in policy documents? Results from an exploratory study*. https://doi.org/https://doi.org/10.6084/m9.figshare.3175168.v1

Levin, N., & Leonelli, S. (2017). How Does One “Open” Science? Questions of Value in Biological Research. *Science Technology and Human Values, 42*(2), 280–305. https://doi.org/10.1177/0162243916672071

Munafò, M. R., Nosek, B. A., Bishop, D. V., Button, K. S., Chambers, C. D., Percie Du Sert, N., … Ioannidis, J. P. (2017). A manifesto for reproducible science. Macmillan Publishers Limited. https://doi.org/10.1038/s41562-016-0021

Open Science Collaboration. (2015). Estimating the reproducibility of psychological science. *Science, 349*(6251). https://doi.org/10.1126/science.aac4716

Peters, I., Kraker, P., Lex, E., Gumpenberger, C., & Gorraiz, J. I. (2017). Zenodo in the Spotlight of Traditional and New Metrics. *Frontiers in Research Metrics and Analytics, 2*(December). https://doi.org/10.3389/frma.2017.00013

Rozin, P. (2009). What Kind of Empirical Research Should We Publish, Fund, and Reward?: A Different Perspective. *Perspectives on Psychological Science, 4*(4), 435–439. https://doi.org/10.1111/j.1745-6924.2009.01151.x

Tukey, J. W. (1980). We need both exploratory and confirmatory. *American Statistician, 34*(1), 23–25. https://doi.org/10.1080/00031305.1980.10482706

Wagenmakers, E. J., Wetzels, R., Borsboom, D., Maas, H. L. van der, & Kievit, R. A. (2012). An Agenda for Purely Confirmatory Research. *Perspectives on Psychological Science, 7*(6), 632–638. https://doi.org/10.1177/1745691612463078

Wagner, C. S., & Alexander, J. (2013). Evaluating transformative research programmes: A case study of the NSF Small Grants for Exploratory Research programme. *Research Evaluation, 22*(3), 187–197. https://doi.org/10.1093/reseval/rvt006

Watson, M. (2015). When will ‘open science’ become simply ‘science’? *Genome Biology, 16*(1), 101. https://doi.org/10.1186/s13059-015-0669-2